# Experimental Investigation on Columns of Steel Fiber Reinforced Concrete with Recycled Aggregates under Large Eccentric Compression Load

**DOI:** 10.3390/ma12030445

**Published:** 2019-01-31

**Authors:** Changyong Li, Haibin Geng, Caiheng Deng, Bingchen Li, Shunbo Zhao

**Affiliations:** 1School of Civil Engineering and Communications, North China University of Water Resources and Electric Power, Zhengzhou 450045, China; lichang@ncwu.edu.cn; 2International Joint Research Lab for Eco-building Materials and Engineering of Henan, North China University of Water Resources and Electric Power, Zhengzhou 450045, China; chdeng@stu.ncwu.edu.cn (C.D.); bc.li@stu.ncwu.edu.cn (B.L.); 3Henan Provincial Collaborative Innovation Center for Water Resources High-efficient Utilization and Support Engineering, Zhengzhou 450046, China

**Keywords:** steel fiber-reinforced concrete with recycled aggregates (SFRC-RA), columns, large eccentric compression load, lateral displacement, crack width, cracking, stiffness

## Abstract

To improve the structural application of steel fiber-reinforced concrete with recycled aggregates (SFRC-RA) composited in gradation by large-particle natural coarse aggregate and small-particle recycled coarse aggregate, the large eccentric compression behavior of eight SFRC-RA columns was experimentally investigated in this paper. The main parameters considered were the strength of the SFRC-RA and the volume fraction of the steel fiber. Details about the sectional concrete strain, the longitudinal steel bar strain, the lateral displacement, the cracking load, the crack distribution and crack width, and the ultimate load of the SFRC-RA columns were measured. The beneficial effects of steel fiber on these attributes were discussed, and the ductility corresponding to the lateral displacement of the SFRC-RA columns was also analyzed. Based on the test results and design principles, formulas were proposed for predicting the cracking resistance, crack width, and lateral displacement of SFRC-RA columns in a normal service state. The ultimate loads of the SFRC-RA columns under a large eccentric compression load were calculated, considering the second-order effects.

## 1. Introduction

With the requirement of sustainable development and environmental protection worldwide, recycled concrete aggregate (RCA) has been commonly used as the new aggregate of concrete in recent years [1,2,3]. According to studies in the literature [4,5,6], the coarse RCA has certain unbeneficial effects on the properties of new concrete, due to the special characteristics of a rough surface, a low density, and a high and quick rate of water absorption that it possesses, which differ from those of natural aggregate. This is rooted in the coarse RCA, attached with a certain amount of old cement mortar. On this point, many technologies were developed for the treatment of coarse RCA, to eliminate the adverse effects. Methods for enhancing the removal of attached old mortar from the coarse RCA were reported, such as the heating and rubbing method and the ultrasonic cleaning technique [7,8]. The approach of coarse RCA surface treatment by means of a surface modifier, such as the microbial carbonate precipitation, the alkaline organosilicone, or the hydrochloric acid plus calcium metasilicate, was demonstrated to be feasible by increasing the weight and reducing the water absorption of treated coarse RCA [9,10,11]. A two-stage mixing method was used to fill the porosity and micro-cracks on the coarse RCA surface at the first stage, by using a stronger cement paste [12,13,14,15] or by pre-coating materials [16,17]. A self-healing process was achieved by immersing the coarse RCA in water for 30 days, to give the unhydrated cement particles attached to the surface of the RCA a good chance to react with water again [14]. However, in respect to getting rid of the old cement mortar from the coarse aggregate, a simple and feasible method is to produce coarse RCA with a maximum particle size smaller than that used in the old concrete [1,2,3,4,18,19,20,21]. To meet the requirement of particle gradation of coarse aggregate, the large-particle natural aggregate and the small-particle recycled aggregate were mixed together to be the composite coarse aggregate [22,23,24]. To further enhance the tensile strength of the composite coarse aggregate, steel fibers can be admixed as the reinforcements [25,26]. Therefore, a novel building material of steel fiber-reinforced concrete with recycled aggregates (SFRC-RA) becomes a reality.

According to the nature of the mechanical properties and failure characteristics, the steel bar at the tension zone of a reinforced concrete column would collapse under a large eccentric compression load [27,28]. This failure pattern has obvious precursors beforehand, which relate to the plastic failure. As reported in previous studies [1,2,29,30,31], the reinforced recycled-aggregate concrete columns always had a lower bearing capacity under large eccentric compression loads compared to the results calculated by using the formulas specified for the design of conventional concrete columns. This is due to the uneliminated adverse effects of coarse RCA on the strength and modulus of the elasticity of concrete, in which the natural coarse aggregate was simply partially replaced by the coarse recycled aggregate. In contrast, the reinforced all-recycled-aggregate concrete columns had an equal or higher bearing capacity compared to the calculated results of the conventional concrete columns with the same strength grade of concrete [32,33], and the cracking resistance and crack width were also equivalent [34]. This demonstrates the importance of the mix proportion design of recycled aggregate concrete, especially for the correct reuse of coarse recycled aggregates in concrete. In order to improve the performance of reinforced recycled concrete columns, an attempt at strengthening the steel fiber was done. The expected results were almost achieved with the great cracking resistance and the small crack width, however, a slightly lower bearing capacity of the columns was obtained [35].

As an example of the structural application of SFRC-RA, the experimental investigation of SFRC-RA columns under a large eccentric compression load was carried out in this paper. Eight reinforced SFRC-RA columns were designed with different strength grades of SFRC-RA and varied volume fractions of steel fiber. The loading behaviors were completely recorded as the foundation of theoretical analyses. The ductility related to the lateral displacement and the sectional flexural stiffness are discussed. The second-order effects on crack extension and the bearing capacity of the testing columns are analyzed. The predictive formulas for the sectional cracking force, crack width, and bearing capacity of the reinforced SFRC-RA columns are proposed.

## 2. Experimental Program

### 2.1. Raw Materials

Crushed limestone, with a particle size of 16–20 mm, and recycled aggregate, with a particle size of 5–16 mm, were mixed in the proportion of 2:3 to make the coarse aggregate in the experiment. The proportion was chosen due to the maximum compact stacking density of the aggregates. A recycled aggregate, with a particle size of 0–5 mm, was used as the fine aggregate. The recycled aggregates were crushed from the tested concrete beams in the lab, while the maximum particle size of the natural aggregate used in the original concrete was 20 mm. The pictures of the fine and coarse aggregates are shown in Figure 1. The distribution of the particle sizes of the aggregates are shown in Figure 2. The physical and mechanical properties of the fine and coarse aggregates is shown in Table 1.

Grade P.O 42.5 ordinary silicate cement was used. Its physical and mechanical properties are listed in Table 2. Polycarboxylic acid superplasticizer of PCA-I, with a water-reducing rate of 30% and a density of 520 kg/m^3^, and tap water were used. Mill-cut steel fiber, with a length of *l*_f_ = 32 mm and an aspect ratio of *l*_f_/*d*_f_ = 40, was used. The picture of the steel fiber is shown in Figure 3.

### 2.2. Preparation of SFRC-RA

Three water-to-cement ratios of 0.35, 0.41, and 0.48 were designed for the SFRC-RA. The absolute volume method was used to calculate the mix proportion of the SFRC-RA [36,37]. Additional water was also added, according to the absorption of the recycled coarse and fine aggregates [21,22,23,24]. The volume fractions of steel fiber (*v*_f_) were 1.2%, 1.6%, and 2.0%, respectively. The sand ratio was 42%. Different dosages of water-reducer were used in the mix proportions of the SFRC-RA to keep the slump in a stable range. The mix proportions and slumps of the SFRC-RA are presented in Table 3.

A forced horizontal shaft mixer was used to produce the SFRC-RA. The fine and coarse recycled aggregates were pre-wet by additional water. The natural aggregate was added and mixed uniformly, then the cement and additive were mixed by adding the mixing water. Finally, the steel fibers were added last.

### 2.3. Design of the SFRC-RA Columns

Eight SFRC-RA columns were designed in this experiment. The cross-section of the columns was 150mm × 300 mm and the length was 2.0 m. The initial eccentric distance (*e*_0_) was 200 mm. The ends of the columns were equipped with brackets and the thickness of the concrete covers (*c*_s_) was 25 mm. In order to prevent local failure on the ends of the columns, steel plates were welded at the ends of the steel bar frames and the stirrups in the brackets were increased. According to the specification in Chinese codes [27,38], the longitudinal steel bars were designed as grade HRB400, with a diameter (*d*) of 16 mm, a measured yield strength (*f*_y_) of 436.9 MPa, and an elastic modulus (*E*_s_) of 2.03 × 10^5^ MPa. The stirrups were grade HRB400, with a diameter of 6 mm. Details of the columns are presented in Figure 4.

As shown in Figure 5, the columns were produced vertically, to be consistent with the actual loading status. The fresh SFRC-RA was poured into a steel form as three layers and was compacted by the slight vibration of the vibrators attached to the outside of the steel form. Six cubes with dimensions of 150 mm were manufactured to measure the cubic compressive strength (*f*_fcu_) and splitting tensile strength (*f*_ft_), while six cylinders of *φ*150 mm × 300 mm were manufactured to measure the cylinder compressive strength (*f*_fc_) and modulus of elasticity (*E*_c_) of the SFRC-RA. All of them were accompanied with the columns in the same batch of the pouring and curing condition. The test results are listed in Table 4.

### 2.4. Test Method

Tests were carried out on a 5000 kN four-column hydraulic testing machine (Changchun New Testing Machine Co., Ltd., Changchun, China). As presented in Figure 4, the hinged supports were coaxially placed on the bottom and top ends of the column. The load was applied by the testing machine on the top end and measured by a 1000 kN loading transducer (East China Electronics Co., Ltd., Yantai, China). Several loading levels were designed according to the measured contents [39]. Five concrete strain gauges were bonded along the mid-height section to verify the section assumption of concrete strain and the reinforcement strain gauges were bonded along the length of the longitudinal steel bars. Five LVDTs (Huayan Electronics Co., Ltd., Wuhan, China) were arranged to measure the lateral displacement of the column along its height. Other measures included the cracking load, the ultimate load, the crack patterns, and the crack width. The crack load was measured according to the strain of the steel bars and the SFRC-CA, and the crack width at the center of the longitudinal tensile rebar was measured by an electrical reading microscope (Zhibo Union Technology Co., Ltd., Beijing, China).

## 3. Results and Discussion

### 3.1. Failure Process

When the columns were loaded to about 15%–20% of the ultimate bearing capacity, transverse micro-cracks appeared on the tensile sides of the columns. With the continuous loading, the number of transverse cracks increased and extended to the compressive sides of the columns. When the loads reached about 60%–70% of the ultimate bearing capacity, the number and lengths of the transverse cracks remained constant, while their widths increased. When the loads reached about 85%–90% of the ultimate bearing capacity, vertical micro-cracks appeared on the compressive side of the columns, and the crack widths and lateral displacements increased rapidly. The ultimate bearing capacity was reached while several vertical macro-cracks at the compressive zone appeared. After that, the bearing capacity of the column was reduced. The failure modes of the experimental columns are shown in Figure 6. It should be noted that the vertically macroscopically cracked SFRC-RA in the compressive zone kept its integrity without peeling off and the number of cracks increased with the volume fraction of the steel fiber.

### 3.2. Concrete Strain of the Mid-height Section

Figure 7 displays the variations in the concrete strain along with the depth of the mid-height section in different load grades for the SFRC-RA columns. Generally, the changes in the concrete strain along the normal section of each column confirms the plane-section assumption [27,28].

### 3.3. Strains of the Longitudinal Steel Bars

Figure 8 presents the curves of the compressive and tensile strains of the longitudinal steel bars. Generally, the steel bars in the compression zone of the SFRC-RA columns were linearly increased with the load. The steel bars in the tension zone of the SFRC-RA columns had two linear increase stages before and after the SFRC-RA cracking, with the increases in load. As the theoretical yield strain of the steel bars was about 2150 *με*, the longitudinal steel bars, under the tension and compression of all of the testing columns, collapsed under the ultimate eccentric loads.

### 3.4. Mid-height Lateral Displacement vs. Load Curves

The mid-height lateral displacement versus the load curves of the experimental SFRC-RA columns is exhibited in Figure 9. Before the cracking of the SFRC-RA, the lateral displacement was relatively small at the initial stage of loading. Once cracks occurred, the lateral displacement gradually increased with loading. When the ultimate loading was reached, the lateral displacement turned into fast growth.

With the increase of the SFRC-RA strength, the slope of the curves before the ultimate loading increased. This means that a higher flexural stiffness was produced due to the higher SFRC-RA strength, which will be discussed in more detail in Section 4.2. When the ultimate loading was reached, the slopes of the curves were almost the same with different SFRC-RA strengths.

With the increasing volume fraction of the steel fiber, the lateral displacements of the columns tended to be smaller before the ultimate loading. This was characterized by the larger slope of the curves and the closer cracks that appeared with smaller widths on both sides of the columns. The presence of steel fibers in the SFRC-RA restricted the growth of cracks, due to the bridging effects across the cracks, and made a new balance of internal forces on the normal cross-section.

### 3.5. Ductility

The displacement ductility coefficient, the ratio of the ultimate displacement to the yield displacement, is always used as an index reflecting the deformability of structural members [28,40]. For simplification, this coefficient (*μ*) is defined as the ratio of the lateral displacement (Δ_85%_) when the residual bearing capacity reaches 85% of the displacement (Δ*_u_*) corresponding to the ultimate bearing capacity:
(1)$$\mu =\frac{\Delta_{85\%}}{\Delta_{u}}$$


The results are presented in Table 5. It can be seen that with the increase in the SFRC-RA strength, the ductility of the column became smaller. This is similar to that of reinforced conventional concrete columns, due to the brittleness of concrete being increased with the strength grade [27,28]. The ductility coefficients of the C40-1.2 and C50-1.2 columns were about 4.5% and 9.5% lower than the C30-1.2 column, respectively. However, the ductility of the columns improved with the increasing volume fraction of steel fiber. The ductility coefficients of the columns C40-1.6 and C40-2.0 were 10.3% and 13.4% higher than that of C40-1.2, respectively. This is due to the benefit from the bridging effects of the steel fibers on the SFRC-RA columns.

## 4. Prediction of the Second-order Effect Factor due to Lateral Displacement

### 4.1. Second-Order Effect

As presented in Figure 10, the lateral displacement of the column under eccentric compression leads to the increase of momentum on the normal section. In the case of the column with equal axial force on the hinged ends, the largest lateral displacement takes place at the mid-height section which is the section under the control of the bearing capacity. Therefore, the momentum due to the increased lateral displacement should be taken into account. The momentum (*M*) on the mid-height section of the column is:
(2)$$M=N(e_{0}+a_{f})$$


Introducing the second-order effect factor (*η*_ns_), it becomes:
(3)$$M=\eta_{\mathrm{ns}}Ne_{0}$$
(4)$$\eta_{\mathrm{ns}}=1+a_{f}/e_{0}$$


The lateral displacement of the SFRC-RA column can be approximately regarded as a sinusoidal curve [27,28], and the equation is obtained according to the differential equation of the displacement curve, of which the boundary conditions are shown in Figure 10:
(5)$$y=a_{f}\sin(\pi x/l_{0})$$
where *y* is a variable of the lateral displacement in SFRC-RA columns of different heights, *x* is an independent variable of the height, and *l*_0_ is the calculated effective height. With an increase in load, *y* increases, and when *x* becomes the mid-height of the SFRC-CA column, *y* becomes the maximum of *a*_f_.

The relationship between the curvature (*φ*) and the displacement (*a*_f_) at the mid-height section is:
(6)$$\phi =\frac{\pi^{2}}{{l_{0}}^{2}}a_{f}$$


Based on the relationship between the curvature (*φ*) and the tensile strain (*ε*_sm_) of the longitudinal tensile steel bars [27,28], it can be written as:
(7)$$\phi =\frac{\varepsilon_{sm}}{\zeta h_{0}}$$
(8)$$\varepsilon_{sm}=\psi \frac{\sigma_{s}}{E_{s}}$$
where *ζ* is the comprehensive coefficient related to the sectional depth of the tension zone, *h*_0_ is the effective sectional depth of the columns, *ψ* is the coefficient related to the deformation of the steel bars, *σ*_s_ is the tensile stress of the longitudinal steel bars calculated in Section 5.2.2, and *E*_s_ is the elastic modulus of the longitudinal reinforcement.

Considering the effect of steel fibers on the stress distribution of the longitudinal tensile steel bars, the tensile strength (f_ft_) of the SFRC-RA is used to replace the tensile strength (*f*_t_) of conventional concrete in the formula of *ψ* [27,28], so:
(9)$$\psi =1.1-0.65\frac{f_{\mathrm{ft}}}{\rho_{\mathrm{te}}\sigma_{s}}$$
where *ρ*_te_ is the effective tensile reinforcement ratio in cross-section. Then, the displacement (*a*_f_) is obtained by substituting Equations (7) and (8) into Equation (6):
(10)$$a_{f}=\frac{\psi \sigma_{s}{l_{0}}^{2}}{\pi^{2}E_{s}h_{0}\zeta }$$


By inputting the test data of this study, the comprehensive coefficient (*ζ*) is 0.33.

By substituting Equation (10) into Equation (4) and using *h*_0_ = 0.87*h*, the second-order effect factor (η_ns_) is obtained under normal service (50%–70%) of the ultimate bearing capacity, and the ultimate bearing capacity is:
(11)$$\eta_{\mathrm{ns}}=1+\frac{\psi \sigma_{s}}{0.33{\left(0.87\pi \right)}^{2}E_{s}}\cdot \frac{1}{e_{0}/h_{0}}\cdot {\left(\frac{l_{0}}{h}\right)}^{2}$$


Table 6 shows the values of the *η*_ns_ for the SFRC-RA columns, along with the lateral displacements under normal service and the ultimate bearing capacity calculated by Equation (10). The ratio of tested-to-calculated lateral displacements is 0.977 on average, with a variation coefficient of 0.068. Good predictive results can be given by these formulas.

To verify the validity of the tests in this paper, Equation (12) and Equation (13) specified in the Chinese design code for concrete structures [27] were used to calculate the second-order effect factor in the crack width and the bearing capacity of the reinforced conventional concrete columns, respectively:
(12)$$\eta_{\mathrm{ns},s}=1+\frac{1}{4000e_{0}/h_{0}}\cdot {\left(\frac{l_{0}}{h}\right)}^{2}$$
(13)$$\eta_{\mathrm{ns},u}=1+\frac{1}{1300e_{0}/h_{0}}\cdot {\left(\frac{l_{0}}{h}\right)}^{2}$$


The value of the *η*_ns,s_ calculated by Equation (12) is 1.015 and the value of the *η*_ns,u_ calculated by Equation (13) is 1.045. The ratio of the *η*_ns,s_ to that of Equation (11), at a normal service state on a loading level from 48% to 71%, is 0.989 on average, with a variation coefficient of 0.006. The ratio of *η*_ns,u_ to that of Equation (11), at an ultimate bearing capacity state on a loading level of 100%, is 0.991 on average, with a variation coefficient of 0.006. This demonstrates that good predictive results can be produced by the two simplified formulas, however, Equation (12) cannot reflect the difference in the *η*_ns,s_ that varied with the loading level.

### 4.2. Lateral Displacement and Flexural Stiffness

Figure 11 displays the measured and predicted values of the lateral displacement along the height of the columns in the normal service state and the ultimate bearing capacity state. It can be seen that the lateral displacement increased due to the decreased sectional flexural stiffness with the increasing load. The assumption of an approximately sinusoidal curve of the lateral displacement, given in Section 4.1, is in good agreement with the measured results.

The lateral displacement of the column under eccentric compression relates to the sectional flexural stiffness. Due to the appearance and extension of cracks, the flexural stiffness varies with the change in the load, which can be obtained from the relationship between the bending momentum, curvature, and stiffness [28]:
(14)$$B_{s}=\frac{M}{\phi }=\frac{\eta_{\mathrm{ns}}Ne_{0}}{\phi }$$


From Equation (10), the flexural stiffness can be calculated as:
(15)$$B_{s}=0.33\frac{\eta_{\mathrm{ns}}Ne_{0}h_{0}E_{s}}{\psi \sigma_{s}}$$


By substituting the measured mid-height displacement into Equation (15), the mid-height sectional flexural stiffness of testing the SFRC-RA columns under a large eccentric compression load was obtained, as presented in Figure 12. It shows that the flexural stiffness of the SFRC-RA columns decreased with the increase in load and that a stronger stiffness of the SFRC-RA columns was exhibited with higher SFRC-RA strength and a larger volume fraction of steel fiber at the same load level.

## 5. Prediction of Cracking

### 5.1. Cracking Resistance

According to the design principle of reinforced concrete structures [28,41], the cracking resistance of the cross-section in the mid-height of SFRC-RA columns is:
(16)$$N_{\mathrm{cr}}=\frac{\gamma_{m}f_{\mathrm{ft}}A_{0}W_{0}}{e_{0}A_{0}-W_{0}}$$
(17)$$\gamma_{m}=1.55(0.73+60/h)$$
(18)$$A_{0}=A_{c}+\alpha_{E}(A_{s}+A_{s}^{\prime })$$
(19)$$W_{0}=I_{0}/(h_{0}-h/2)$$
(20)$$I_{0}=bh^{3}/12+\alpha_{E}(A_{s}+A_{s}^{\prime }){\left(h_{0}-h/2\right)}^{2}$$
where *N*_cr_ is the cracking force of the SFRC-RA column, *γ*_m_ is the plastic influence coefficient of the resistance momentum of the concrete section, *A*_0_ is the area of the cross-section, *A*_c_ is the area of the concrete section, *W*_0_ is the ratio of *I*_0_ to the distance from the edge of the tension zone to the section centroid, *I*_0_ is the momentum of the inertia of *A*_0_ to the section centroid, *α*_E_ is the elastic modulus ratio, *α*_E_ = *E*_s_/*E*_c_, *E*_c_ is the elastic modulus of the concrete listed in Table 4, *b* is the sectional width, and *h* is the sectional depth.

The mean ratio of the tested-to-calculated values of the cracking load is 1.059, with a variation coefficient of 0.033, as listed in Table 7. The predictions are in good agreement with the test results.

### 5.2. Crack Width

#### 5.2.1. Average Crack Spacing

Based on the bond stress-slip theory [27,28,41] of the crack width, the average crack spacing along the height of the SFRC-RA columns can be calculated by the following formulas:
(21)$$l_{m}=k_{1}c_{s}+k_{2}\frac{d}{\rho_{\mathrm{te}}(1+\alpha_{t}\lambda_{f})}$$
(22)$$\rho_{\mathrm{te}}=\frac{A_{s}}{0.5bh}$$
where *l*_m_ is the average crack spacing, *k*_1_ is a coefficient related to the SFRC-RA cover for the longitudinal tensile steel bars, *k*_2_ is a coefficient related to the bond property between the SFRC-RA and steel bars, *ρ*_te_ is the effective tensile reinforcement ratio in cross-section, α_t_ is a strengthening coefficient of the bond property due to the presence of steel fibers, and *λ*_f_ is the fiber factor (*λ*_f_ = *v*_f_·*l*_f_/*d*_f_).

By the fitting analysis of the average crack spacing in this study, the results of *k*_1_ = 1.86, *k*_2_ = 0.10, and α_t_ = 0.37 can be obtained. For the average crack spacing of the reinforced conventional concrete column specified in Chinese code [27,28], values of *k*_1_ = 1.9 and *k*_2_ = 0.08 can be obtained. Therefore, to link this with the current specification, the formula of the average crack spacing is expressed as follows:
(23)$$l_{m}=1.9c_{s}+0.08\frac{d}{\rho_{\mathrm{te}}(1+0.37\lambda_{f})}$$


The calculated values of the average crack spacing are compared to the tested values, as presented in Table 8. The mean ratio of the tested-to-calculated results is 1.024 on average, with a variation coefficient of 0.012. Good predictions can be obtained with formula (23).

#### 5.2.2. Average Crack Width

By introducing the coefficient (α_cf_) reflecting the tensile deformation of the SFRC-RA and the coefficient (*ψ*) representing the non-uniform strain of the steel bars within the average crack spacing, and based on the bond stress-slip theory of crack width, the formula [27,28] for calculating the average crack width of reinforced SFRC-RA columns can be expressed as:
(24)$$w_{m}=\alpha_{\mathrm{cf}}\psi \frac{\sigma_{s}}{E_{s}}l_{m}$$
where *w*_m_ is the average crack width.

Considering the effects of steel fibers on the average crack width, the α_cf_ value will be reduced with an increase in the fiber factor. Equation (24) can be rewritten as:
(25)$$w_{m}=0.77(1-\beta_{w}\lambda_{f})\psi \frac{\sigma_{s}}{E_{s}}l_{m}$$
where 0.77 is the coefficient of the conventional concrete specified in Chinese code (α_cf_) [27,28] and *β*_w_ is a coefficient related to the presence of steel fibers.

The stress of the tensile longitudinal steel bars at the cracked section can be calculated with the equilibrium of axial forces, considering the effect of steel fibers in the tension zone. As seen from Figure 13, the following formula can be obtained by the equilibrium momentum at the resultant point of the compression zone:
(26)$$\sigma_{s}=\frac{N(e-z)-\sigma_{\mathrm{sf}}bx_{t}(z-x_{t}/2+a_{s})}{A_{s}z}$$
(27)$$z=[0.87-0.12{\left(\frac{h_{0}}{e}\right)}^{2}]h_{0}$$
(28)$$e=\eta_{\mathrm{ns},s}e_{0}+h/2-a_{s}$$
(29)$$\sigma_{\mathrm{sf}}=\alpha_{t}\lambda_{f}f_{t}$$
(30)$$x_{t}=0.5h$$
where *e* is the distance between the loading position and the resultant position of the steel bars in the tension zone, *z* is the distance from the resultant point of the steel bars in the tension zone to the resultant point of the compression zone, *σ*_sf_ is the tensile stress of steel fibers in the crack section, *α*_t_ is the influence coefficient of steel fibers on the tensile strength of the SFRC-RA, *f*_t_ is the tensile strength of the SFRC-RA base without steel fibers (obtained from *f*_ft_ = *f*_t_(1 + *α*_t_*λ*_f_)), and *x*_t_ is the effective depth of the tension zone influenced by steel fibers.

By fitting the average crack width of the testing columns in this study, it can be deduced that *β*_w_ = 0.19 and *α*_t_ = 0.37. By using the above formulas, the average crack width of each testing column can be calculated, the results of which are presented in Table 9. The ratio of tested-to-calculated values is 0.967 on average, with a variation coefficient of 0.047. As a result, the average crack width decreased with the increase of the volume ratio of steel fiber. Good predictions can be obtained from Equation (25).

#### 5.2.3. Maximum Crack Width

The maximum crack width (*ω*_max_) can be obtained by multiplying an enlargement factor (*α*_s_) by the average crack width (*ω*_max_) [27,28]. By using a histogram analysis of the distribution [28,34,35] of the crack widths (*ω*_i_), as presented in Figure 14, the ratio *ω*_i_/*ω*_m_ demonstrates a normal distribution, with an average value of 0.934 and a standard deviation of 0.405. With a 95% reliable probability, an enlargement factor (*α*_s_ = *μ* + 1.645*σ* = 1.60) can be calculated. This value is slightly smaller than the value of 1.66 for reinforced conventional concrete columns. Due to the lack of large amounts of test data about SFRC-RA columns, the same value of 1.66 for reinforced conventional concrete columns is used in this paper. Therefore, the formula for calculating the maximum crack width of SFRC-RA columns under a large eccentric compression load is as follows:
(31)$$w_{\max}=1.66w_{m}$$


By using Equation (31), the maximum crack widths of SFRC-RA columns in a normal service state are calculated, with the results presented in Table 9. The ratio of the tested-to-calculated maximum crack width is 0.957 on average, with a variation coefficient of 0.170.

## 6. Prediction of Bearing Capacity

Based on the equilibrium of forces on the mid-height section [38], as presented in Figure 15, the formulas for calculating the bearing capacity (*N*_u_) of SFRC-RA columns under large eccentric compression loads are given out as below:
(32)$$N_{u}=\alpha_{1}f_{\mathrm{fc}}bx_{c}-\sigma_{sf}bx_{\mathrm{ft}}+0.87f_{y}^{\prime }A_{s}^{\prime }-f_{y}A_{s}$$
(33)$$N_{u}e=\alpha_{1}f_{\mathrm{fc}}bx_{c}(h_{0}-x_{c}/2)+0.87f_{y}^{\prime }A_{s}^{\prime }(h_{0}-a_{s}^{\prime })-\sigma_{sf}bx_{\mathrm{ft}}(x_{\mathrm{ft}}/2-a_{s})$$
(34)$$x_{\mathrm{ft}}=h-x_{c}/\beta_{1}$$
(35)$$e=\eta_{\mathrm{ns},u}e_{0}+h/2-a_{s}$$
where *α*_1_ is the coefficient influenced by the compressive strength, *f*_y_′ is the yield strength of the compressive reinforcement bars, *x*_c_ is the depth of the compressive zone of concrete, *x*_ft_ is the depth of the tensile zone of concrete, *A*_s_′ is the section area of the compressive reinforcement bar, *a*_s_ is the distance from the resultant point of the tensile bar to the edge of the tensile zone, *a*_s_′ is the distance from the resultant point of the compressive bar to the edge of the compressive zone, and *β*_1_ is the coefficient related to the depth of compression.

In accordance with Chinese code [27,38], *α*_1_ = 1.0 and *β*_1_ = 0.8. By substituting the measured axial compressive strength (*f*_fc_) and the splitting tensile strength (*f*_ft_) of the SFRC-RA, as well as the tested yield strength (*f*_y_ and *f*_y_′) of the longitudinal steel bars into the above equations, the ultimate axial force of the SFRC-RA columns are able to be obtained, as listed in Table 10. The ratio of the tested-to-calculated values is 0.942 on average, with a variation coefficient of 0.061. Good agreement is achieved by using these formulas for the calculation of ultimate axial force.

## 7. Results and Conclusions

Based on the experimental research of the behavior of SFRC-RA columns under large eccentric compression loads, the following conclusions can be drawn:

(1) The normal section of the SFRC-RA columns confirms the plane-section assumption during the loading process up to the ultimate state. The cross-section close to the mid-height of the columns fails, with vertical macroscopic cracks in the SFRC-RA compressive zone. A higher bearing capability after the ultimate load leads to a better ductility of the SFRC-RA columns with large lateral displacement, and the ductility increases with an increase in the volume fraction of steel fiber.

(2) The increase in the cracking resistance of SFRC-RA columns is related directly to the tensile strength of the SFRC-RA. The formulas for reinforced conventional concrete columns can be used for reinforced SFRC-RA columns, by substituting in the corresponding tensile strength (*f*_ft_) of the SFRC-RA.

(3) Due to the presence of steel fibers, the crack spacing decreased with the uniform distribution of the SFRC-RA strains, and the crack width decreased with the lower tensile stress of the longitudinal steel bars. Based on the bond stress-slip theory and the principle linking reinforced conventional concrete columns, the predictive formulas of crack spacing, average crack width, and maximum crack width are proposed.

(4) The bearing capacity of the testing columns was obviously contributed to by the SFRC-RA strength, while a certain increment was benefited by the steel fibers. Considering the beneficial effect of steel fibers in the tensile zone of the controlled section, the formulas for predicting the axial force at the bearing capacity state are suggested. The acceptable accuracy of the predictions of the major design characteristics indicates the suitability of the SFRC-RA for its structural application.

## Figures and Tables

**Figure 1 materials-12-00445-f001:**
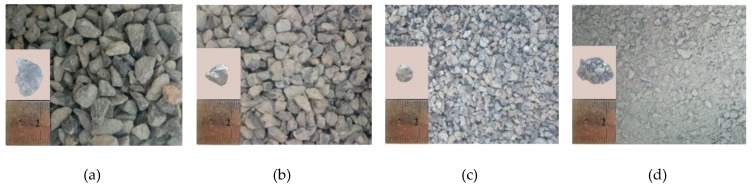
The fine and coarse aggregates: (**a**) The natural aggregate of 16–20 mm, (**b**) The recycled aggregate of 10–16 mm, (**c**) The recycled aggregate of 5–10 mm, and (**d**) The recycled aggregate of 0–5 mm.

**Figure 2 materials-12-00445-f002:**
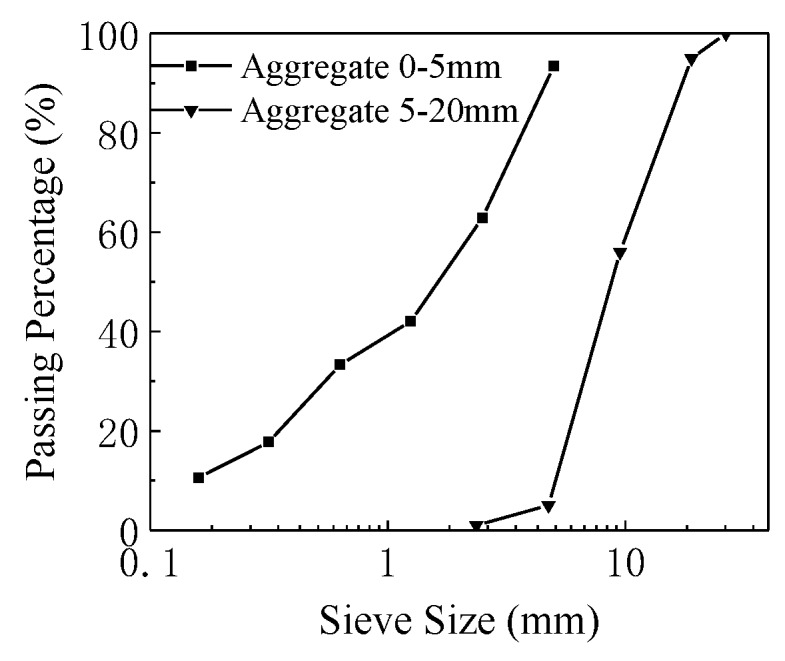
The particle size distribution of the aggregates.

**Figure 3 materials-12-00445-f003:**
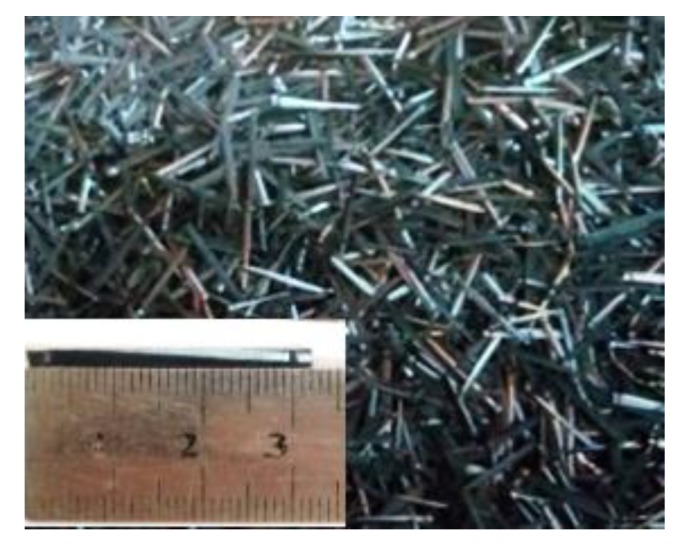
The mill-cut steel fibers.

**Figure 4 materials-12-00445-f004:**
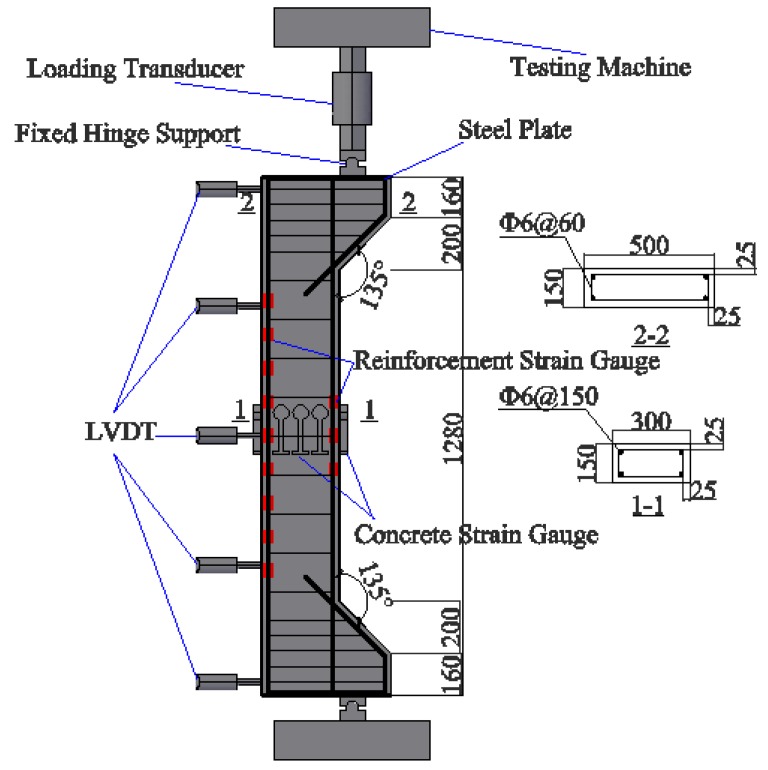
The details of the specimens and the layout of the measurements.

**Figure 5 materials-12-00445-f005:**
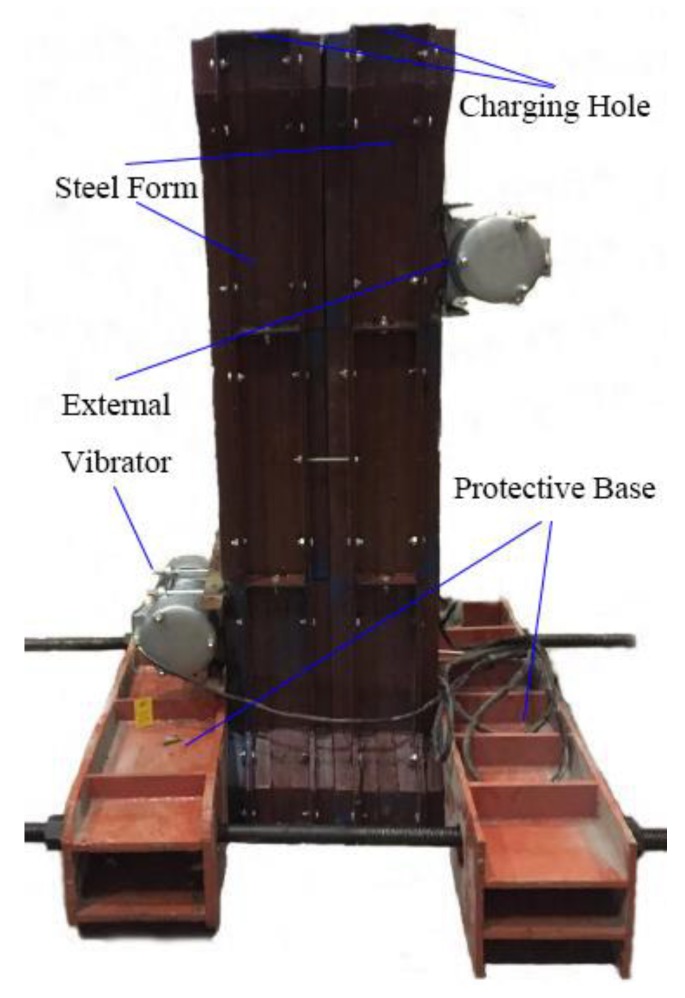
The pouring method and the equipment.

**Figure 6 materials-12-00445-f006:**
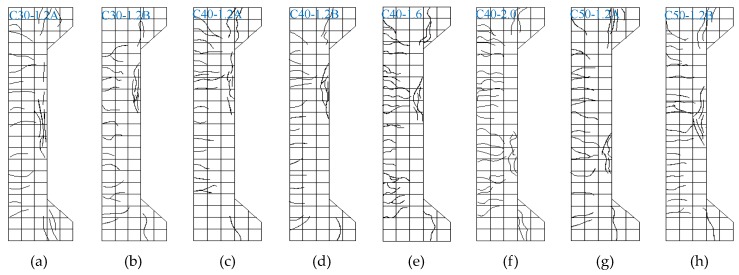
The failure modes of the columns: (**a**) C30-1.2A; (**b**) C30-1.2B; (**c**) C40-1.2A; (**d**) C40-1.2B; (**e**) C40-1.6; (**f**) C40-2.0; (**g**) C50-1.2A; (**h**) C50-1.2B.

**Figure 7 materials-12-00445-f007:**
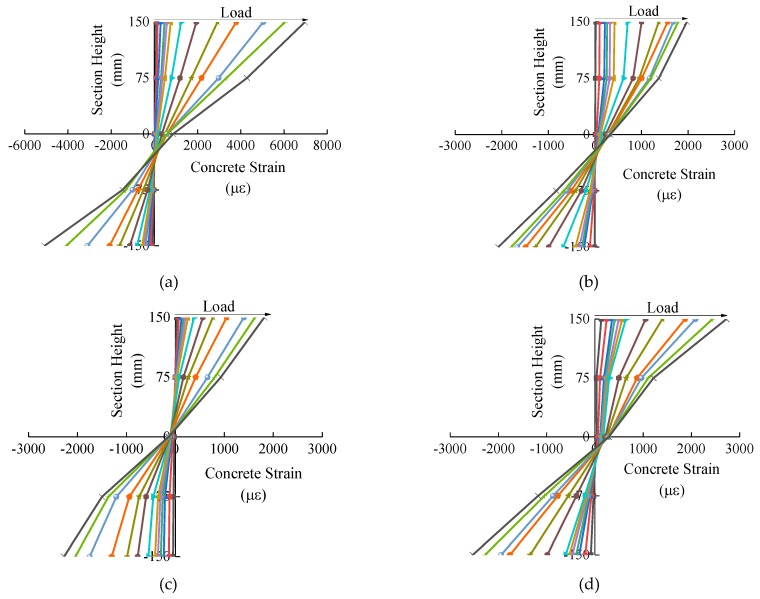
The strain curves of the normal section of the columns: (**a**) C30-1.2A; (**b**) C30-1.2B; (**c**) C40-1.2A; (**d**) C40-1.2B; (**e**) C50-1.2A; (**f**) C50-1.2B; (**g**) C40-1.6; (**h**) C40-2.0.

**Figure 8 materials-12-00445-f008:**
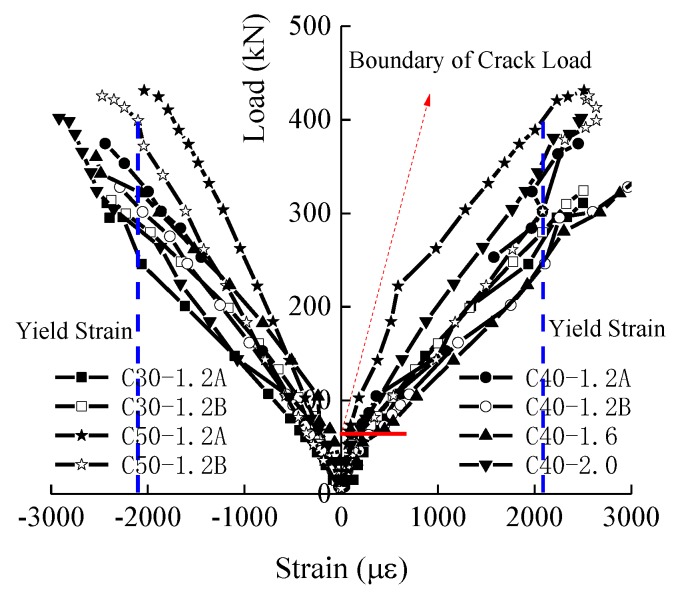
The load-reinforced steel strain curves.

**Figure 9 materials-12-00445-f009:**
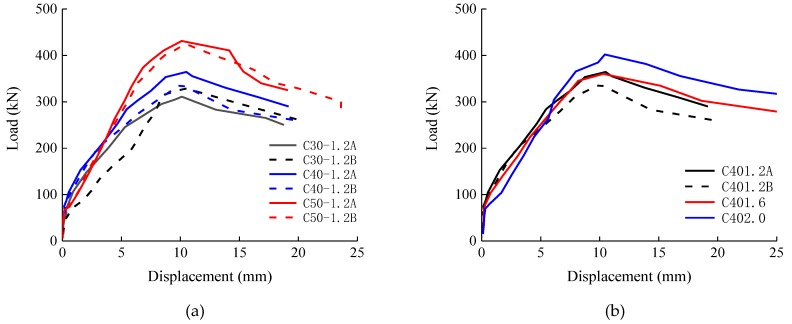
The mid-height lateral displacement versus load curves: (**a**) Concrete strength; (**b**) Volume fraction of steel fiber.

**Figure 10 materials-12-00445-f010:**
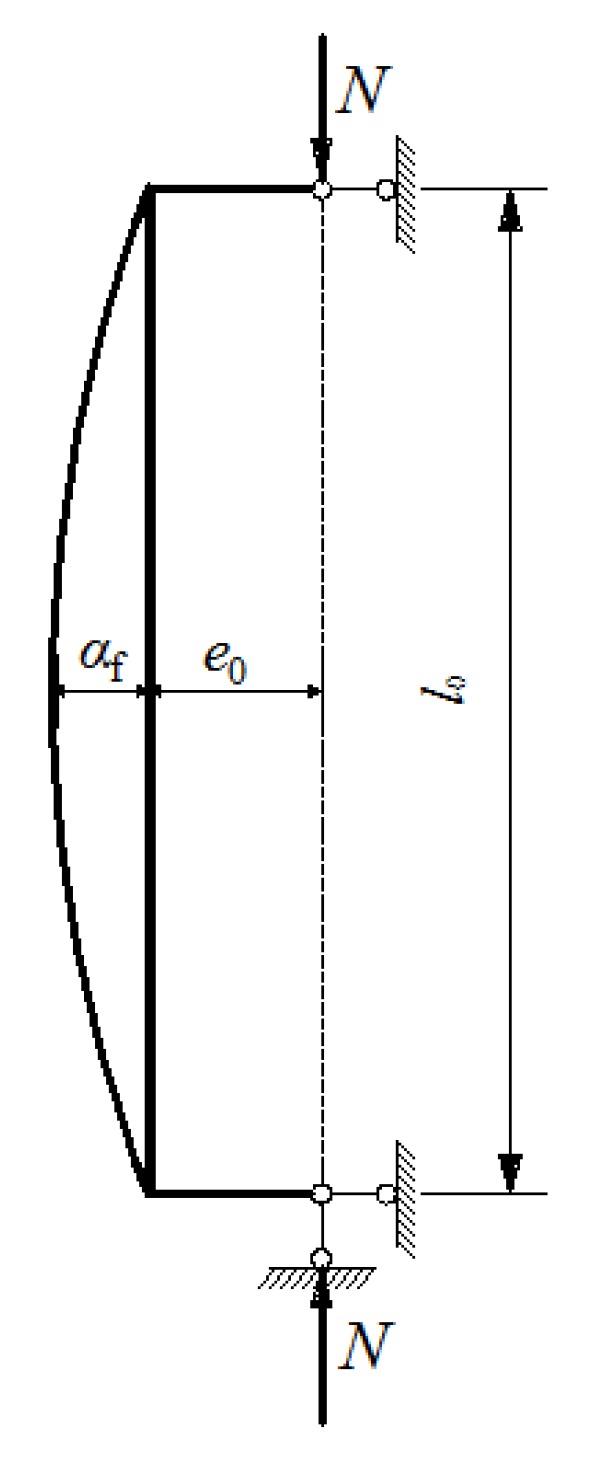
The longitudinal bending diagram of the second-order effect.

**Figure 11 materials-12-00445-f011:**
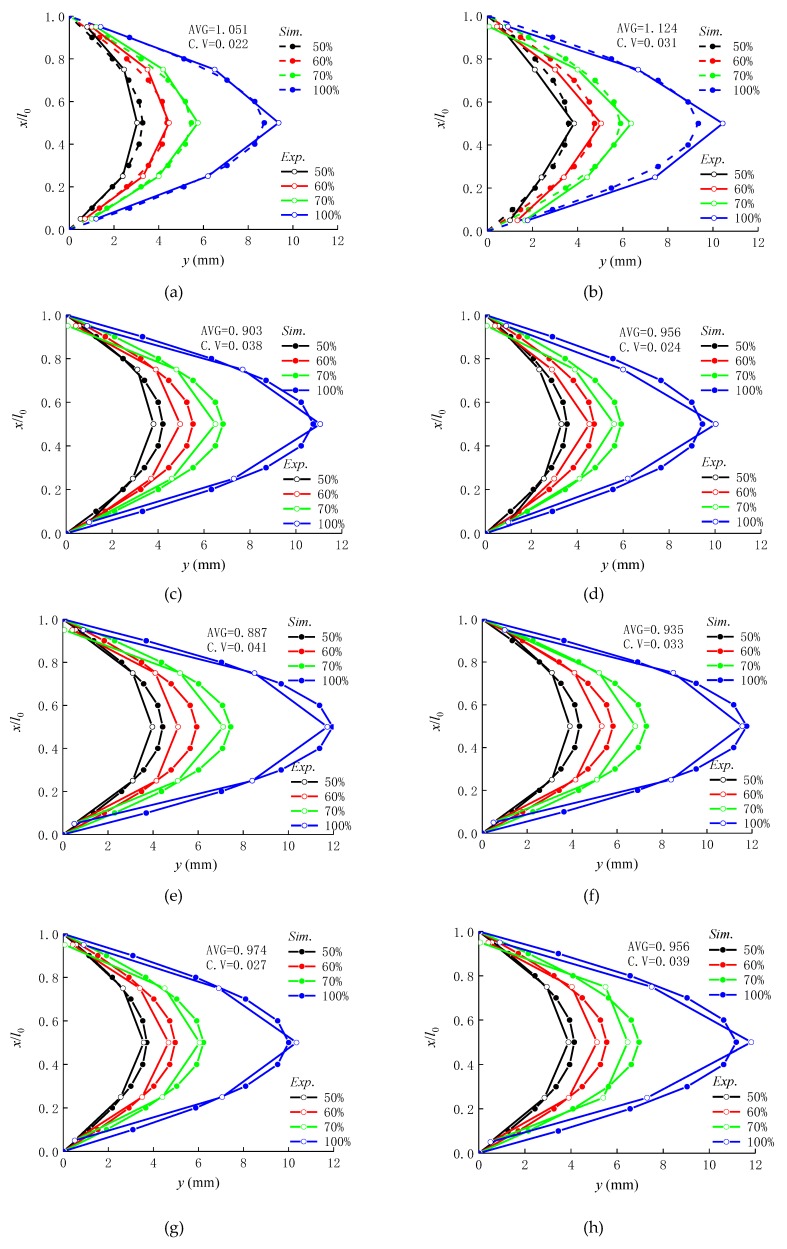
The displacement curves along the height of the columns: (**a**) C30-1.2A; (**b**) C30-1.2B; (**c**) C40-1.2A; (**d**) C40-1.2B; (**e**) C50-1.2A; (**f**) C50-1.2B; (**g**) C40-1.6; (**h**) C40-2.0.

**Figure 12 materials-12-00445-f012:**
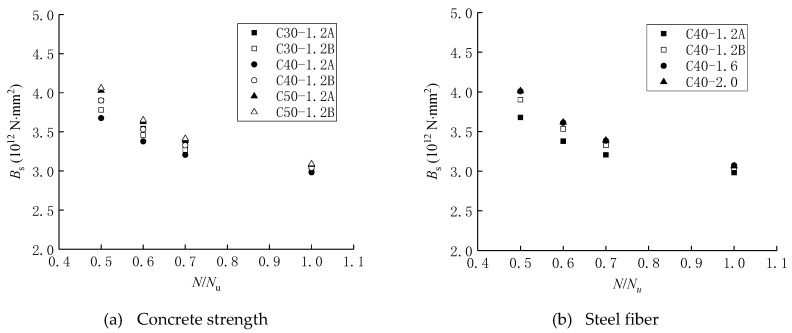
The flexural stiffness: (**a**) Concrete strength; (**b**) Volume fraction of steel fiber.

**Figure 13 materials-12-00445-f013:**
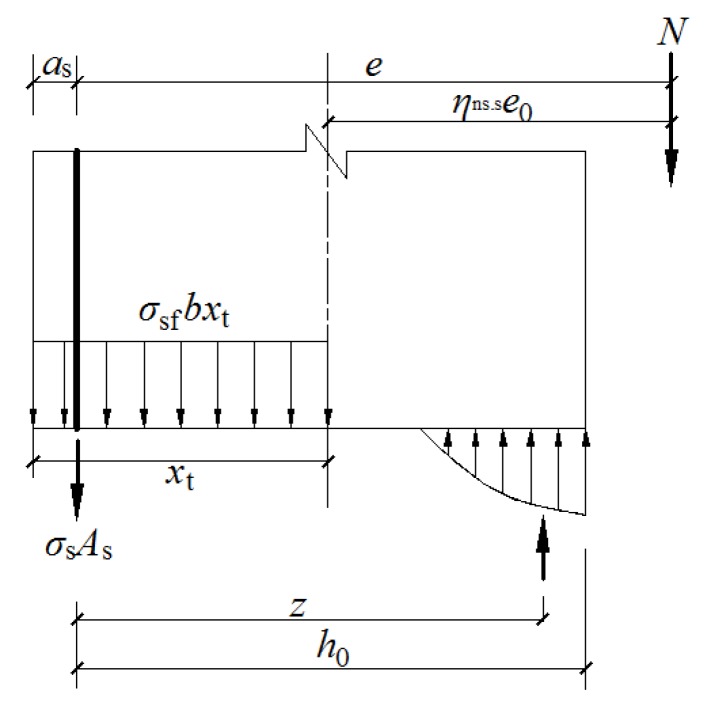
The equilibrium of forces on the normal section of the column.

**Figure 14 materials-12-00445-f014:**
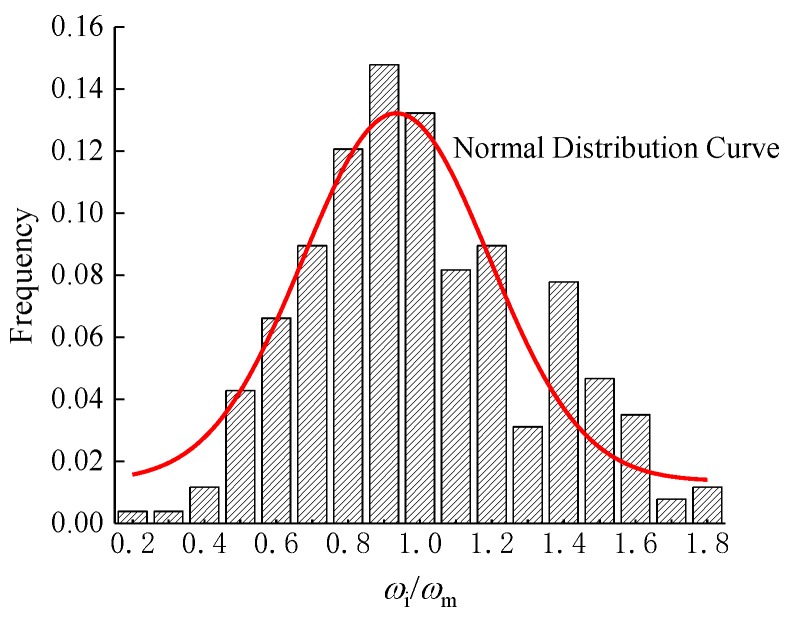
A histogram of the tested crack width distribution.

**Figure 15 materials-12-00445-f015:**
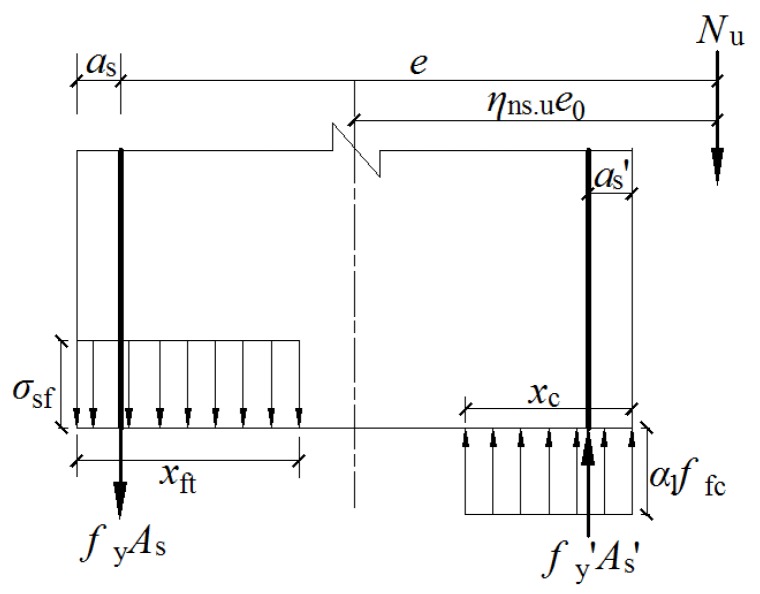
The bearing capacity under a large eccentric compression load on a normal section of the column.

**Table 1 materials-12-00445-t001:** The physical and mechanical properties of the fine and coarse aggregates.

Properties	Coarse Aggregate		Fine Aggregate
	Natural	Recycled	
Apparent Density (kg/m^3^)	2720	2690	2440
Bulk Density (kg/m^3^)	1520	1350	1260
Compact Stacking Density (kg/m^3^)	1670	1450	1470
Water Absorption at 24 Hours (%)	0.9	5.9	8.5
Crush Index (%)	14.1	10.0	14.1
Fineness Modulus	-	-	3.22

**Table 2 materials-12-00445-t002:** The physical and mechanical properties of cement.

Grade	Density (kg/m^3^)	Consistency	Setting Time (min)		Compressive Strength (MPa)		Flexural Strength (MPa)	
			Initial	Final	3 d	28d	3 d	28d
42.5	3050	26.9	168	269	28.9	45.2	4.0	5.3

**Table 3 materials-12-00445-t003:** The mix proportion of the steel fiber-reinforced concrete with recycled aggregates (SFRC-RA) (kg/m^3^).

Mix ID	C30-1.2	C40-1.2	C40-1.6	C40-2.0	C50-1.2
***w*/*c***	0.48	0.41	0.41	0.41	0.35
Water	180	180	180	180	180
Cement	373.2	437.1	437.1	437.1	510
Recycled Fine Aggregate	738.5	710	718.3	726.6	677.6
Recycled Coarse Aggregate	555.4	531.8	519.8	507.8	505
Natural Coarse Aggregate	370.2	354.5	346.5	338.6	336.6
Steel Fiber	94.2	94.2	125.6	157	94.2
Additional Water	66.3	63.6	63.7	63.8	60.6
Water-Reducer	2.24	3.06	3.50	3.93	4.08
Slump (mm)	150	150	140	130	150

**Table 4 materials-12-00445-t004:** The tested basic mechanical properties of the steel fiber-reinforced concrete with recycled aggregates (SFRC-RA).

Specimens ID	ƒ_fcu_ (MPa)	ƒ_fc_ (MPa)	ƒ_ft_ (MPa)	*E*_c_/10^4^ MPa
C30-1.2A/B	30.0	22.6	1.97	2.83
C40-1.2A/B	38.3	28.8	2.13	2.98
C50-1.2A/B	48.5	38.4	2.84	3.38
C40-1.6	40.4	31.2	2.25	3.05
C40-2.0	41.6	33.8	2.41	3.11

**Table 5 materials-12-00445-t005:** The ductility coefficient (*μ*) of the tested columns.

Specimens	C30-1.2A	C30-1.2B	C40-1.2A	C40-1.2B	C50-1.2A	C50-1.2B	C40-1.6	C40-2.0
***μ***	1.70	1.66	1.58	1.63	1.52	1.54	1.77	1.82

**Table 6 materials-12-00445-t006:** The second-order effect factor and lateral displacement.

Specimens	*N*/*N*_u_	*η* _ns_	*a*_f_ (mm)		
			Test	Calculation	Test/Calculation
C30-1.2A	48%	1.016	3.0	3.2	0.936
	58%	1.021	4.5	4.3	1.041
	71%	1.027	5.8	5.3	1.078
	100%	1.045	9.4	9.1	1.033
C30-1.2B	52%	1.018	3.8	3.5	1.084
	61%	1.023	5.0	4.7	1.083
	70%	1.029	6.4	5.8	1.105
	100%	1.048	10.4	9.7	1.074
C40-1.2A	48%	1.021	3.8	4.1	0.918
	59%	1.027	5.0	5.4	0.918
	69%	1.033	6.3	6.7	0.941
	100%	1.056	11.1	11.2	0.992
C40-1.2B	50%	1.017	3.2	3.5	0.915
	59%	1.023	4.2	4.6	0.910
	71%	1.029	5.6	5.8	0.970
	100%	1.049	11.0	9.8	1.123
C50-1.2A	51%	1.022	4.0	4.4	0.908
	60%	1.029	5.3	5.8	0.911
	70%	1.036	6.8	7.3	0.935
	100%	1.062	11.7	12.5	0.942
C50-1.2B	47%	1.021	3.9	4.3	0.912
	59%	1.028	5.2	5.7	0.914
	71%	1.036	6.5	7.1	0.911
	100%	1.061	11.6	12.2	0.945
C40-1.6	50%	1.018	3.6	3.7	0.974
	61%	1.024	4.7	4.9	0.957
	69%	1.030	6.1	6.1	0.993
	100%	1.052	10.4	10.4	0.994
C40-2.0	50%	1.020	3.9	4.1	0.949
	62%	1.027	5.1	5.5	0.941
	70%	1.034	6.5	6.8	0.949
	100%	1.058	11.8	11.6	1.016

**Table 7 materials-12-00445-t007:** The tested and calculated values of the cracking loads.

Specimens		C30-12A	C30-12B	C40-12A	C40-12B	C50-12A	C50-12B	C40-16	C40-20
*N*_cr_ (kN)	Tested	48.1	45.8	50.1	53.5	63.4	65.3	55.2	58.3
	Calculated	44.9	44.9	48.0	48.0	62.6	62.6	50.5	53.9
	Tested/Calculated	1.071	1.021	1.043	1.113	1.012	1.042	1.092	1.081

**Table 8 materials-12-00445-t008:** The calculated and tested values of the average crack spacing (*l*_m_) of the test columns.

Specimens		C30-12A	C30-12B	C40-12A	C40-12B	C50-12A	C50-12B	C40-1.6	C40-2.0
*l*_m_ (mm)	Tested	123	121	121	125	124	121	119	117
	Calculated	120	120	120	120	120	120	117	114
	Tested/Calculated	1.028	1.012	1.014	1.047	1.033	1.013	1.020	1.023

**Table 9 materials-12-00445-t009:** The calculated and tested values of the average (*w*_m_) and maximum crack width (*w*_max_) of the tested columns.

Specimens	*N*/*N*u	*w*_m_ (mm)			*w*_max_ (mm)		
		Calculated	Tested	Tested/Calculated	Calculated	Tested	Tested/Calculated
C30-1.2A	48%	0.06	0.07	1.109	0.10	0.12	1.233
	58%	0.08	0.08	1.030	0.13	0.15	1.163
	71%	0.10	0.10	1.034	0.16	0.20	1.245
C30-1.2B	52%	0.06	0.06	0.950	0.10	0.13	1.239
	61%	0.08	0.08	0.963	0.14	0.16	1.160
	70%	0.10	0.10	0.972	0.17	0.19	1.112
C40-1.2A	48%	0.07	0.07	0.946	0.12	0.13	1.059
	59%	0.10	0.09	0.931	0.16	0.16	0.997
	69%	0.12	0.12	1.006	0.20	0.18	0.909
C40-1.2B	50%	0.06	0.06	0.927	0.11	0.07	0.652
	59%	0.09	0.08	0.933	0.14	0.12	0.843
	71%	0.11	0.10	0.936	0.18	0.15	0.846
C50-1.2A	51%	0.08	0.08	1.005	0.13	0.10	0.757
	60%	0.11	0.10	0.942	0.18	0.15	0.851
	70%	0.13	0.13	0.979	0.22	0.19	0.862
C50-1.2B	47%	0.08	0.07	0.918	0.13	0.11	0.869
	59%	0.10	0.10	0.981	0.17	0.15	0.886
	71%	0.13	0.12	0.940	0.21	0.19	0.896
C40-1.6	50%	0.06	0.06	0.966	0.10	0.09	0.873
	61%	0.08	0.08	0.966	0.14	0.11	0.800
	69%	0.10	0.10	0.966	0.17	0.14	0.815
C40-2.0	50%	0.07	0.07	0.989	0.11	0.11	1.008
	62%	0.09	0.08	0.914	0.15	0.14	0.963
	70%	0.11	0.10	0.914	0.18	0.17	0.936

**Table 10 materials-12-00445-t010:** The tested and calculated ultimate force of the test columns.

Specimens		C30-12A	C30-12B	C40-12A	C40-12B	C50-12A	C50-12B	C40-1.6	C40-2.0
*N*_u_ (kN)	Tested	311.1	329.2	374.5	337.2	431.3	425.5	360.1	402.1
	Calculated	356.7	356.7	386.3	386.3	424.1	424.1	398.4	411.2
	Tested/Calculated	0.872	0.923	0.970	0.873	1.017	1.003	0.904	0.978

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
