# Peer review of "Experimental Investigation on Columns of Steel Fiber Reinforced Concrete with Recycled Aggregates under Large Eccentric Compression Load"

_materials, 2019, doi:10.3390/ma12030445_

Round 1

Reviewer 1 Report

Having both valuable and weak points, the submission makes a controversial impression on the reviewer. On the one hand, the reported experimental results can be interesting for potential readers of the Journal. On the other hand, however, a scientific background of the study is missing. Novelty of the research is unclear. The experimental results are described practically without any comments and explanation of physical nature of the observed effects. The writing style is unacceptable as well. This manuscript can be considered for publication in Materials only after major modification. The following comments must be addressed:

General comment. The acronym “SFR-HRAC” is not suitable. The acronym “SFRC”(= steel fiber reinforced concrete) is well known. In this study, concrete aggregates were partially replaced with recycled material. Hence, the considered concrete should be designated as “SFRC-RA”(= steel fiber reinforced concrete with recycled aggregates).

1) Research object:

The study is not motivated. What is the main difference of the present study concerning the previous works reported in literature (e.g., references [22]-[30])?

The research object must be related with material aspects of the concrete with recycled aggregates and steel fiber reinforcement. The authors must keep in minds that consideration of the structural element different from considered in previous works cannot be considered as a sufficient scientific value. The columns should be considered only as example of structural application of SFRC-RA.

The Chinese Design Code (GB50010-2010) was used as the reference for efficiency analysis of SFRC-RA: the acceptable accuracy of the predictions of major design characteristics indirectly indicates suitability of the considered concretes for the structural application. That is acceptable, but this idea must be clearly described in the manuscript.

Several coefficients were proposed for the Design Code models representing the mechanical properties of SFRC-RA. However, not any comments were made on the effect of fiber content and the aggregates replacement ratio on value of the coefficients.

2) Prediction verification. The aforementioned coefficients were arbitrarily set as to represent the experiential results adequately. The prediction results indicate adequacy of this procedure, but do not guarantee accuracy of the Design Code models. In other words, the same data was used for modification of the Code models and verification of the prediction that is unacceptable: the coefficients are suitable for modelling the considered specimens only. Accuracy of the predictions must be verified by using alternative experimental results.

3) International auditory of the Journal. Design principles by the Code GB50010 should be compared with prediction models described in world acknowledged international standards (either ACI or Model Code). In this concern, the reviewer would also like to point out unacceptability of the statement (Introduction) “With the requirement of national green and sustainable development…” What is the situation in other countries? How this study contributes to the international researches?

4) Conclusions:

Conclusion 1. The sentence “fails with the crushed SFR-HRAC and yield of longitudinal steel bars” contradicts to the description given in Section 3.1. The primary failure cause must be identified. Is it related with fiber content?

Conclusion 2 is not novel.

Conclusion 3. The statement “by substituting corresponded tensile strength of concrete” is too abstract. The fiber effect must be quantified.

The title of the section should be changed to “Results and Conclusions” since Conclusion 4 is a result, but not a conclusion.

5) Writing style. The text contains a lot of typo and other inaccuracies. Due to a multiple number of such occurrences, complete list of errors is not included in this report. A native speaker should carefully revise the text by improving it stylistically. This is a strict requirement for further consideration of this work. A wrong terminology is also used (e.g., “abandoned (?) concrete beams”, “original (?) natural aggregate”, “lime stone”, “axial (cylinder?) compressive strength”, “partial compression (local failure?)”, “To consist with the actual loading status (?), the columns were produced in vertical (position?)”, “load was exerted (?)”, “measured by a loading transducer (?)”, “grades (?) of load”, “Plane Assumption (?)”, “lateral deflection (displacement?)”). The terms “hybrid” and “green” are uninformative and incorrect in the present context (e.g., “The hybrid-recycled-aggregate was composited…”, “were hybrid together”). The personal writing style (referring by a writer to himself via “we”, “our”, etc.) is also inappropriate.

6) Other comments and suggestions:

Title. The term “large eccentric compression” requires a clarification. What is the load condition of the “large” eccentricity? This condition must be clarified in the text. In any case, “large eccentric compression” should not be identified as the research object (see the second paragraph of Comment 1 of this report, above).

Abstract and Introduction must be rewritten as stylistically imperfect and unclear. They must describe scientific novelty of the research and motivate the study.

Section 2.1 must be rewritten as stylistically imperfect.

Sections 2.1 and 2.2. Material choice must be substantiated properly since the referred literature sources [38, 39] are not accessible in English. In particular, it is unclear why limestone aggregates were used since low strength and low deformation modulus are      characteristic of this material. It is known that quality of recycled aggregates varies significantly. Mechanical properties of the recycled aggregates (e.g., content of the attached mortar) should be characterized. The volume content of fibers (1.2%...2.0%) also seems very high. The choice of such high fiber content must be substantiated. The maximum      aggregate size (20 mm) is non-proportionally big for the considered fiber contents. How was uniformity of fiber distribution in the concrete achieved/verified? Table 1 must indicate the additional water content and the actual slump.

Line 129. How was strain of the bar reinforcement measured?

Line 149. The wording “the load until yielding” is misleading. Failure of the specimens was associated with the crushing of the      compressive concrete (Section 3.1). The corresponding deformation stages in Figure 7 must be indicated. The authors should keep in mind that different mechanisms govern failure of bar reinforcement in tension and in compression.

Section 3.4 (and throughout the text).Lateral displacements” must be referred instead of the “lateral deflections”. This section must be rewritten as stylistically imperfect. For instance, the sentences “After cracked (?), the lateral deflection gradually increased”, “This means that a higher flexural stiffness was given out due to the higher SFR-HRAC      strength (what is the reference?)”, and “The presence of steel fibers in SFR-HRAC confined (?) the growth of crack…” are incorrect and uninformative. The corresponding deformation/failure mechanisms must be described.

Section 3.5. The assumption “For the simplification (?), this coefficient μ is defined as the ratio of the lateral deflection Δ85% when the residual bearing capacity reaches 85% ultimate to the deflection Δu corresponded to the ultimate bearing capacity”      requires a clarification. Why is this assumption acceptable? An illustrative scheme is necessary for the statement “…with the increase of SFR-HRAC strength, the ductility of the column became smaller”. The statement “This benefited from the confinement effects (?) of steel fibers on the integrity (?) of SFR-HRAC in compression zone” is misleading.

Section 4.1. The terms “an independent variable of height” and “comprehensive coefficient” are misleading. They must be described. The statement “Then, the formula (?) is obtained as follow” is unclear. Figure 9 must indicate the support (boundary) conditions. How were the experimental displacements obtained in Table 4? The statement “under normal service and ultimate states” is unclear. What are the corresponding loading conditions? The same comment is related with Section 4.2 (“…in the normal service state and the peak-load state”). The sentence “To simplify the calculation process…” is unclear. What is the problem? Seemingly, it is related with the improper description of the research object (see the third paragraph of Comment 1, above). The Design Code GB50010 should be introduced as the reference estimating structural efficiency of the SFRC-RA. The similarity of the coefficients identified using the test results and specified in the Code indirectly indicates suitability of the concrete for the structural application.

Section 4.2. The sentences “the assumption of approximately sinusoidal curve of lateral deflection is suitable in good agreement with the measured results” and “The lateral deflection of column under eccentric compression relates to the sectional flexural stiffness” must be linked to appropriate test results. How were the “sinusoidal curves” calculated? The wording “the higher stiffness of SFR-HRAC columns existed (?) with higher SFR-HRAC strength and larger volume fraction of steel fiber” is stylistically imperfect and unclear. What are the mechanical properties responsible for the flexural stiffness increase? What is the effect of the aggregates replacement ratio?

Section 5.1. The sentence “According to the design principle of reinforced concrete structure (structures?) [44, 45], and considering the large strain gradient and sufficient plastic deformation in the tension zone (where is it shown?) of SFR-HRAC      columns under large eccentric compression” requires a clarification. The terms “controlled section”, “elastic resistance moment”, and “conversion section” are not commonly used. How was the elasticity modulus of concrete, Ec, determined? The wording “It is good in agreement between tested (test) and calculated results” is stylistically imperfect (“results” were not “tested” – “predictions can be in a good agreement with test results”) and must be clarified. How was the crack resistance determined experimentally? The authors must keep in minds that accuracy of results of visual inspection might be unacceptably low for the comparative analysis.

Section 5.2.1. A literature source must be referred in the statement “Based on the bond-slip (bond stress-slip?) theory [?]” The wording “The ratio of tested to calculated values” is incorrect as well as the term “Good predictive result”.

Section 5.2.2. The statements “The average crack width is equal to the difference between the tensile length (?) of steel bars and SFR-HRAC within the average crack spacing” and “Differ from the tensile deformation of conventional concrete within average crack spacing, the tensile deformation of SFR-HRAC is restrained (?) by the steel fibers crossing cracks” are wrong. Literature reference is necessary for Equation (25). The term “loading point” is unacceptable in the text on Lines 289-293.

Section 5.2.3. The cracking analysis (so called “histogram analysis”) technique is interesting. It must be described in detail. Was it reported somewhere? How was the histogram (Figure 13) constructed? Why is the normal probability distribution law used to approximate this histogram? The statement “The maximum crack width ωmax is always (not always) obtained by multiplying an enlargement factor αs on the average crack width ωm” is wrong in general (e.g., the average crack width is not used in the Model Code 2010). The term “quality heterogeneity” is unsuitable. Table 7 should be referred on Line 312.

Section 6.1 should not be presented as a separate section (Section 6 contains only one sub-section). The analysis assumptions should be checked: failure of compressive steel is related with buckling that is realized at the stresses which absolute magnitude is below the yield strength (it can be assumed approximately equal to 0.87fy). The sentence on Line 323 should not begin with a symbol.

Author Response

Thanks for the reviewer's careful revision.A response to the comments was uploaded as a Word.

Reviewer 2 Report

This is an actual, informative and interesting manuscript. The preparing is accurate but can find some small editing mistakes.  

The abstract and the conclusions were writing according to submitted research material. 

Maybe the notations part is not necessary, because used symbols are explained directly in the text.

Author Response

Dear Professor,

Thanks very much for your careful review and kindly reminder. It is my honor to receive your review report. I look forward to the next cooperation. Thank you again.

Sincerely,

Haibin Geng

Reviewer 3 Report

This manuscript analyses experimentally the eccentric compression behavior of eight columns made of green hybrid-recycled-aggregate concrete reinforced with steel fibers (SFR-HRAC). The paper may be valuable. However it needs major revision at this stage. Here are some guidances: 

- in the Introduction I suggest to discuss single references when cited. For example it does not much information to the reader that it is written that "according to the studies in literatures [1-6]" or [12-16] in line 45. Maybe some references are reduntant and should be deleted as unappropiate?

- p. 2.1. Because the article focus on the use of recycle aggregate it will be beneficial to provide the particle size distribution (sieve size and granular analysis) and the chemical composition of used aggregate. It will allow other researchers to validate the study,

- line 84. Please provide a reason why the mix proportion of 5-16mm recycled aggregate to 20mm natural aggregate was selected to be 3:2. It will be beneficial to provide a preliminary analysis of this selection,

- p. 2.1. Please provide the procedure of the production of coarse recycled aggregate. What was the source of the aggregate? How was it prepared? What was the chemical composition?

- line 87. Please provide more details about the cement. What was the class of it? It will be beneficial to provide the chemical composition and basic properties of cement. What was the specific surface area, setting time and loss of ignition?

- line 88. What was the density of the plasticizer? Please provide a manufacturer and type of the plasticizer,

- line  89. Please provide more propoerties of fibres,

- line 91. Please provide the standard applied to obtain the designed strength grade of C30, C40 and C50. These are very stange types because most of the standards use C30/37 instead of C30,

- line 92. Please justify why the selected volume ratios for steel fibres was 1.2, 1.6 and 2.0%. How was this volume ratio calculated? Is it really possible to do it?

- line 93. Authors wrote that the workability of fresh mixture was about 150mm slump. How did it changed as a function of the dosage of steel fibres? This information should be provided,

- Figures 1 and 2 do not add much to the content of the paper because it presents only visual observations,

- p. 2.3. How the stell bars influenced the obtained results? Please merge fig. 3 and 4 into one figure,

- p. 3.1. Please provide more scientific analysis of the cracks observed on the surface. It is not enough to provide only a photos after failure. How the cracks propagated? Please draw these cracks. What were the differences between the samples? After observing these photos (Fig. 5) I do not see any differences,

- overall the manuscript is too long. Please concider to merge some figures and delete some text. The manuscript will look more significant if it will be reduced to maximum of 20 pages.

Author Response

Thanks for the reviewer's careful revision. A point-by-point response to the comments was uploaded as a Word.

Round 2

Reviewer 1 Report

The manuscript was modified in an acceptable manner; the reply by the authors is also thorough. This manuscript is recommended for publication in Materials. The reviewer suggest the following minor modifications (no re-revision is necessary) that will help improving quality of the presentation:

Line 233. A symbol should not begin a sentence.

Lines 245-246. The sentence “The assumption of approximately sinusoidal curve (?) of lateral displacement is suitable in good agreement with the measured results” is remained      unclear. Please, clarify the sinusoidal approximation technique in the text in the same manner as it was done in the reply-letter (i.e. “The sinusoidal curves can be calculated according to differential equation of the displacement curve”). How was the differential      equation defined?

Lines 313-314. Please, use less categorical statement “can be” (instead of “is always”) in the statement “The maximum crack width ωmax is always (can be) obtained by multiplying an enlargement factor αs on the average crack width ωm”.

Author Response

Point 1:Line 233. A symbol should not begin a sentence.

Response 1: Thanks for your kind reminder. It has been revised.

Point 2:The sentence “The assumption of approximately sinusoidal curve (?) of lateral displacement is suitable in good agreement with the measured results” is remained      unclear. Please, clarify the sinusoidal approximation technique in the text in the same manner as it was done in the reply-letter (i.e. “The sinusoidal curves can be calculated according to differential equation of the displacement curve”). How was the differential equation defined?

Response 2:Thanks for your kind reminder. It has been revised as "the assumption of approximately sinusoidal curve proposed in Section 4.1..." "the equation is abtained according to differential equation of the displacement curve, of which the boundary conditions are shown in Fig 10.".

Point 3:Please, use less categorical statement “can be” (instead of “is always”) in the statement “The maximum crack width ωmax is always (can be) obtained by multiplying an enlargement factor αs on the average crack width ωm”.

Response 3: Thanks for your kind reminder. It has been revised.

Reviewer 3 Report

All changes were considered. Thus, I suggest to accept this manuscript for publication

Author Response

Thanks for your kind suggestion. It's my pleasure to accept your advice.